# Evolution of the *Ace-1* and *Gste2* Mutations and Their Potential Impact on the Use of Carbamate and Organophosphates in IRS for Controlling *Anopheles gambiae s.l.*, the Major Malaria Mosquito in Senegal

**DOI:** 10.3390/pathogens11091021

**Published:** 2022-09-07

**Authors:** Moussa Diallo, Ebrima SM Kolley, Abdoulaye Kane Dia, Mary Aigbiremo Oboh, Fatoumata Seck, Jarra Manneh, Abdul Karim Sesay, Seynabou Macote Diédhiou, Pape Cheikh Sarr, Ousmane Sy, Badara Samb, Oumar Gaye, Ousmane Faye, Lassana Konaté, Benoit Sessinou Assogba, El Hadji Amadou Niang

**Affiliations:** 1Disease Control and Elimination Theme, Medical Research Council Unit, The Gambia at the London School of Hygiene and Tropical Medicine, Banjul P.O. Box 273, The Gambia; 2Laboratoire d’Ecologie Vectorielle et Parasitaire, Faculté des Sciences et Technique, Université Cheikh Anta Diop, Dakar 10200, Senegal; 3Genomics Core Platform, Medical Research Council Unit, The Gambia at the London School of Hygiene and Tropical Medicine, Banjul P.O. Box 273, The Gambia; 4Laboratoire de Parasitologie Médicale, Pharmacie et d’Odonto-Stomatologie, Faculté de Médecine, Université Cheikh Anta Diop, Dakar 10200, Senegal

**Keywords:** insecticide resistance, *Anopheles gambiae s.l.*, *Ace-1*, *Gste2*, evolution, genetic diversity, haplotypes, Senegal

## Abstract

Widespread of insecticide resistance amongst the species of the *Anopheles gambiae* complex continues to threaten vector control in Senegal. In this study, we investigated the presence and evolution of the *Ace-1* and *Gste2* resistance genes in natural populations of *Anopheles gambiae s.l.,* the main malaria vector in Senegal. Using historical samples collected from ten sentinel health districts, this study focused on three different years (2013, 2017, and 2018) marking the periods of shift between the main public health insecticides families (pyrethroids, carbamates, organophosphates) used in IRS to track back the evolutionary history of the resistance mutations on the *Ace-1* and *Gste2* loci. The results revealed the presence of four members of the *Anopheles gambiae* complex, with the predominance of *An. arabiensis* followed by *An. gambiae*, *An. coluzzii*, and *An. gambiae–coluzzii* hybrids. The *Ace-1* mutation was only detected in *An. gambiae* and *An*. *gambiae–coluzzii* hybrids at low frequencies varying between 0.006 and 0.02, while the *Gste2* mutation was found in all the species with a frequency ranging between 0.02 and 0.25. The *Ace-1* and *Gste2* genes were highly diversified with twenty-two and thirty-one different haplotypes, respectively. The neutrality tests on each gene indicated a negative Tajima’s D, suggesting the abundance of rare alleles. The presence and spread of the *Ace-1* and *Gste2* resistance mutations represent a serious threat to of the effectiveness and the sustainability of IRS-based interventions using carbamates or organophosphates to manage the widespread pyrethroids resistance in Senegal. These data are of the highest importance to support the NMCP for evidence-based vector control interventions selection and targeting.

## 1. Introduction 

The fight against malaria has made significant progress over the recent decades toward eliminating the disease [1]. This achievement was made possible by the scale-up of effective insecticide-based vector control interventions, such as long-lasting insecticidal nets (LLINs) and indoor residual spraying (IRS), which led to the decrease of overall malaria cases by 70% [2,3]. However, the spread of insecticide resistance within the natural populations of major malaria vector to almost all the public health insecticides, resulting from widespread and uncontrolled use of pesticides both in the public health and agriculture sectors, threaten the sustainability of the gain and sustainability of current and future insecticide-based vector control interventions. However, the threat from increasing insecticide resistance can be effectively managed through a good insecticide management programme and the implementation of targeted and evidence-based vector control strategies. The two main resistance mechanisms underlying the insecticides resistance described so far involve target-site mutations and over-expression of metabolic enzymes detoxifying the insecticides. 

The overexpression of the *Gste2* family group of enzyme (Glutathione S-Transferase Epsilon 2) has been associated with organophosphate- and organochlorine-resistant insects strains [4,5]. This enzyme family has recently been implicated in the resistance of *An. gambiae* to pyrethroids [6]. Moreover, previous studies have reported 5–8-fold *Gste2* over-expression in *An. gambiae* displaying phenotypic DDT resistance, while the voltage-gated sodium channel mutation was not detected [7,8]. The *Gste2*-I*114T* mutation (inducing the overexpression of the *Gste2*) associated with DDT and pyrethroid resistance in African malaria vectors has been reported in Benin, Burkina, Ghana, and Gambia [6,9,10,11]. In addition, reduced susceptibility of *An. gambiae s.l.* to organophosphate and carbamate insecticides was associated with a single amino-acid substitution of the glycine (GGC) by serine (AGC) at codon 119 (G119S) in the *Ace-1* gene (Acetylcholinesterase 1) [12] and was reported from several West African countries, including Senegal, Benin, Burkina Faso, and Ivory Coast [13,14,15,16]. Moreover, at *Ace-1* locus, there are homogeneous-duplication- and heterogeneous-duplications-resistant alleles, which enhances the resistance level against organophosphate-carbamate insecticides [17]. Both the above-mentioned mutations (G119S*-Ace-1* and *Gste2*-I*114T*) were associated with organophosphate-carbamates cross-resistance currently used for IRS in many Sub-Saharan African countries, including Senegal.

To sustain standard LLINs (long-lasting insecticidal nets) only threated with pyrethroids, the World Health Organization (WHO) recommends the application of insecticides with different classes of insecticide with different modes of action mainly through IRS [3]. Therefore, the interest in using organophosphate for public health purposes increased [13,18,19] with widespread reports of their effectiveness against pyrethroid-resistant *Anopheles gambiae* mosquitoes [20,21] and the beneficial impact of IRS with organophosphate in many African countries [22,23,24,25]. However, a concomitant presence of the *Ace-1* and the *Gste2* resistant alleles in the wild *Anopheles* populations might jeopardize the use of organophosphate insecticides for IRS to manage the wide spread of pyrethroids resistance.

In this study, we tracked back the presence and evolution of *Ace-1* and *Gste2* resistance genes in several *Anopheles gambiae s.l.* wild populations collected at different time points corresponding to shift periods from pyrethroids to carbamate and then organophosphate used for IRS in the selected health districts of Senegal.

## 2. Results

### 2.1. Ace-1 and Gste2 Resistant Allele Frequencies in Anopheles gambiae s.l. from Senegal

A total of 1647 samples were screened for insecticide-resistance markers at *Ace-1* and *Gste2* loci from wild *An. gambiae s.s.*, *An. coluzzii*, *An. arabiensis*, and *An. gambiae-coluzzii* hybrids populations across all the study sites. 

The G119S mutation at *Ace-1* locus was only found in *An. gambiae s.s.* and *An. gambiae-coluzzii* hybrids in the health districts of Velingara, Tambacounda, and Kedougou. The resistant allele frequency of the G119S mutation was 0.02, 0.03, and 0.06, respectively, in the *An. gambiae s.s.* population from Velingara, Tambacounda, and Kedougou. However, this frequency in *An. gambiae s.s.* was 0.04, 0.01, and 0.03, respectively, for the years 2013, 2017, and 2018, without a significant difference (*p* = 0.36). Moreover, in *An. gambiae-coluzzii* hybrids, the mutation was only found in Tambacounda in 2013, with an allele frequency of 0.13 (see Table 1 and Table 2).

The *Gste2*-I*114T* mutation was found in all four *An. gambiae s.l.* species (*An. arabiensis*, *An. gambiae s.s.*, *An. coluzzii*, and *An. gambiae-coluzzii* hybrids). However, the mutation was the most frequent within the natural *An. coluzzii* population and found in eight study sites (except Ndoffane and Tambacounda) with a frequency varying between 0.14 in Koungheul and 0.58 in Velingara (see Table 3). The *Gste2*-I*114T* mutation was found in only three out of the ten study health districts for both *An. arabiensis* (allele frequency of 0.03, 0.03, and 0.18, respectively, from Tambacounda, Ndoffane, and Velingara) and *An. gambiae s.s.* (allele frequency of 0.16, 0.21, and 0.11, respectively, from Kedougou, Tambacounda, and Velingara). It was also found in *An. gambiae-coluzzii* hybrids only in Velingara and Nioro, with frequencies of 0.20 and 0.50, respectively.

Overall, the mean allele frequency of the *Gste2*-I*114T* allele was 0.02 ± 0.003 in *An. arabiensis*, 0.13 ± 0.15 in *An. gambiae s.s.*, 0.25 ± 0.07 in *An. coluzzii*, and 0.20 ± 0.17 in *An. gambiae-coluzzii* hybrid (see Table 3). The mean allele frequency of *Gste2*-*114T* mutation was significantly higher in *An. coluzzii* and *An. gambiae-coluzzii* hybrid than in the other species (one-way ANOVA F = 5.14, *p* = 0.006). In addition, the mean of resistant allele frequency was 0.13, 0.06, and 0.1, respectively, for the years 2013, 2017, and 2018, without a clear temporal trend (one-way ANOVA F = 0.9, *p* = 0.47) as well as for species (see Table 4).

### 2.2. High Genetic and Haplotype Diversities at the Ace-1 Locus in Anopheles gambiae s.l. from Senegal

Thirty-one samples, collected in 2017 and 2018, were successfully sequenced at the *Ace-1* locus. The polymorphism of the partial sequence of 535 bp segment of the *Ace-1* was analysed. The result revealed no gap, 518 monomorphic sites, and 17 polymorphic sites. Amongst the polymorphic sites, five were singletons with two-variable sites (at the positions 330, 333, 339, 369, and 513), while twelve were parsimony with eleven two-variable sites (at the positions 18, 25, 126, 129, 135, 144, 161, 240, 282, 367, and 495) and one three-variable site (at the position 321). The overall DNA polymorphism analysis with DnaSP revealed 22 haplotypes with a high haplotypic diversity of 0.908 and a low nucleotide diversity of 0.00474. Neutrality tests indicated a negative Tajima’s D = (−1.02860), suggesting the abundance of rare alleles with a recent selective sweep removing variation. In addition, the genetic differentiation between 2017 and 2018 was not significant (Chi2 = 25.054; *p*-value = 0.2448; df = 21) with strong hybridization between the two study years (Fst = 0.0053). However, a significant genetic differentiation was observed between all the study sites (Chi2 = 221.880; *p*-value = 0.05; df = 189) with a moderate gene flow (Fst = 0.13970) and between the three species (Chi2 = 73.558; *p*-value = 0.0019; df = 42) but with a limited gene flow (Fst = 0.158).

Furthermore, the signature of a putative selection was investigated to know the main haplotypes and their distribution amongst the *An. gambiae s.l.* species using Templeton, Crandall, and Sing (TCS) methods [26] in DnaSP and Network softwares. The haplotypes H2, H1 (found in all three species), H6, H12 (only detected in *An. coluzzii* and *An. gambiae s.s.*), and H14 (only present in *An. arabiensis*) were the most frequent (Figure 1). The haplotypes H8, H9, and H22 were specific to *An. gambiae s.s.* and were the ones carrying the resistant allele at the *Ace-1* gene (see Figure 1). Moreover, the haplotype diversity was high in all three species, with 0.92 in *An. arabiensis*, 0.78 in *An. coluzzii*, and 0.88 in *An. gambiae s.s.* (13, 8, and 7 haplotypes, respectively, Figure 1). Finally, the phylogenetic relationship inferred with the neighbour-joining method in MEGA revealed the presence of two main clades, with the major clade divided into two sub-clades (one consisting only of *An. arabiensis* and the second with all the species) and the second clade constituted only of *An. coluzzii* and *An. gambiae s.s.* (Figure 2).

### 2.3. High Genetic and Haplotype Diversities at Gste2 Locus in Anopheles gambiae s.l. from Senegal

Twenty-four samples, collected in 2017 and 2018, were successfully sequenced for the *Gste2* locus. The polymorphism analysis of the partial sequence of 459 bp showed 406 monomorphic sites, 4 gaps or missing data, and 49 variable or polymorphic sites. Amongst the variable sites, 27 were variable singletons, with 26 carrying two-variable sites and 1 harbouring three-variable sites, while 22 were parsimony informative sites (20 with two-variable sites and two with three-variable sites). The overall DNA polymorphism analysis revealed the presence of 31 haplotypes with a high haplotype diversity of 0.951 and a nucleotide diversity of 0.01661. Neutrality tests indicated a negative Tajima’s D (−1.23482), suggesting the abundance of rare alleles with a recent selective sweep. In addition, the genetic differentiation between 2017 and 2018 was not significant (Chi2 = 32.336; *p*-value = 0.35; df = 30), with strong hybridization between both years (Fst = −0.00197). Similarly, the genetic differentiation between all the study sites was not significant (Chi2 = 264.667; *p*-value = 0.1314; df = 240), while a significant gene flow (Fst = 0.249) was recorded between the sites (Fst = 0.165) and the three species (Chi2 = 75.674; *p*-value = 0.0835; df = 60).

In addition, the haplotype network was inferred to know the major haplotypes and their distribution amongst *An. gambiae s.l.* species using Templeton, Crandall, and Sing (TCS) methods [26] in DnaSP and Network softwares and revealed that the haplotype H1 is ubiquitous (found in all three species), while the H10 haplotype was only found in *An. arabiensis* and *An. gambiae s.s.* These two haplotypes were the most frequent (Figure 3). However, the haplotype H29 was only present in *An. gambiae s.s.*, harbouring the homozygous resistant allele. Meanwhile the haplotype H1 (presents in all three species); the haplotypes H2, H3, H4, H15, H16, and H23 (only found in *An. coluzzii*); and the haplotypes H7, H30, and H31 (present only in *An. arabiensis*) were harbouring the heterozygous resistant allele. The remaining haplotypes harboured the homozygous susceptible genotype. The haplotypic diversity was high in all three species: 0.95 in *An. arabiensis*, 0.87 *An. coluzzii*, and 0.95 in *An. gambiae s.s.* (14, 12, and 8 haplotypes, respectively, see Figure 3), suggesting that *Gste2* locus may also be a reservoir of novel genetic variants.

Moreover, the phylogenetic relationship inferred with the neighbour-joining method in MEGA revealed the presence of two main clades: the major clade is formed by all species, and a second clade is constituted only of *An*. *arabiensis* and *An. gambiae s.s.* (Figure 4).

## 3. Discussion

Insecticide resistance is a serious threat to vector control and could potentially jeopardize the malaria control gain obtained since 2000. Therefore, monitoring the evolution of genes involved in resistance is a major asset in insecticide-resistance management. Here, we studied the evolution of the mutations in the *Ace-1* and *Gste2* genes in *An. gambiae s.l.* populations collected from ten health districts of Senegal over three years, marking the main period of shift of insecticide molecules families used in IRS in Senegal from pyrethroids to carbamates and then organophosphates used in IRS. Indeed, as in several other African countries, carbamate and organophosphate are used as alternative chemical families for IRS to control malaria in areas where high pyrethroid resistance is confirmed [18,27]. However, given the increasing reports of the resistance to the two later insecticides amongst wild malaria vector populations across the African continent, it is critical to fully characterize all the underlying resistance mechanisms to preserve the limited vector control tools and avoid operational failure of carbamate/organophosphate-based interventions. In Senegal, targeted IRS has been implemented since 2007 in selected health districts through the PMI’s financial and technical support. The entomological surveillance to monitor the entomological parameters, including insecticide resistance, revealed resistance to carbamate in several *An. gambiae s.l.* populations [28,29].

This study revealed the presence of the *Ace-1* mutation in *An. gambiae s.l.* population from the surveyed area of Senegal, thus confirming reports from previous studies in the country [16,30]. However, over the study period and area, the mutation was only found in *Anopheles gambiae s.s* and *An. gambiae-coluzzii* hybrids. The observed low frequency of the resistant allele might be related to the high fitness cost of the G119S-*Ace-1* mutation [31]. The G119S mutation at *Ace-1* locus was widely reported in several African countries, including Ghana, Gambia, Ivory Coast, Burkina Faso, and Cameroon [10,32,33,34,35]. The origin of a mutation could be either by a *de novo* apparition amongst local *An. gambiae s.s.* populations or a genetic migration from neighbouring populations as previously described [35]. Our haplotype network result (see Figure 1) strongly supports the *de novo* apparition hypothesis. However, the haplotypes H8, H9, and H22 carrying the mutation were peripheral and likely evolved from the older and interior haplotype H2.

Moreover, agricultural selection pressure has been documented for carbamate resistance in *An. gambiae* populations across many West African countries [14,18]. Such a selection factor of the insecticide-resistance mechanism might be responsible for the observed G119S-*Ace-1* mutation selection in Senegal, where carbamates are being used in agricultural activities [36]. In addition, the G119S*-Ace-1* mutation induces resistance against organophosphates [12], which have been used recurrently in several IRS sites in Senegal. However, Senegal started implementing IRS in 2007 with financial and technical support from the U.S. President’s Malaria Initiative. IRS was conducted with different pyrethroid insecticides and formulations between 2007 and 2011. Then, the NMCP shifted to bendiocarb between 2011 and 2014 [37], following the evidence of increasing phenotypical resistance to pyrethroids, correlated with the increase of both the East and West kdr mutations. Finally, due to the low residual efficacy of bendiocarb, organophosphates were implemented since 2014 to replace it [38]. Both selection pressures (from agriculture and malaria mosquito control intervention) might have likely selected the G119S*-Ace-1* mutation selection, which was found at low frequency across all three years of study. In this study, the G119S*-Ace-1* mutation was first detected in 2013, and its frequency remained stable over the three selected years. The selection of G119S*-Ace-1* mutation was likely through the use of the carbamate in the agricultural sector and then strengthened during the successive shift to carbamate and organophosphates for IRS implementation. Indeed, the G119S*-Ace-1* mutation was mainly found in *An. gambiae s.s.* (see Table 1). This result suggests a species dependency linked with its bio-ecology favouring the selection of *Ace-1* (G119S) mutation. However, in contrast with *An. coluzzii*, *An. gambiae s.s.* oviposit in temporary, small, rain-filled breeding sites [39]. Then, residual carbamate and organophosphate insecticides from pest control could quickly drain into the breeding site and contribute to the selection of resistant alleles at the *Ace-1* locus.

In addition, the *Ace-1* (G119S) mutation was identified as a heterozygote in most of the samples, suggesting the presence of heterogeneous duplication associated with low fitness cost [40]. However, we did not identify the known mixture of two nucleotides at a single position in those heterozygote samples sequenced except the G119S mutation one, which might suggest the presence of heterogeneous duplication [17]. Therefore, the samples carrying the mutation are the standard heterozygote with susceptible and resistant alleles of *Ace-1* gene. Knowing that the resistant allele of *Ace-1* gene is associated with deleterious fitness cost [17] strongly supports its low frequency observed in Senegal.

The result of the *Gste2* genotyping showed the presence of the mutation in the studied *An. gambiae s.l.* populations for all the three years of study. To our best knowledge, this is the first report of the single mutation point conferring the metabolic resistance amongst wild *An. gambiae s.l.* in Senegal although it has been reported elsewhere in natural *An. gambiae s.l.* populations [7,9] and *An. funestus* [41,42,43]. Indeed, the mutation was widely reported in several African countries, such as the Gambia [32], Ghana [10], Benin [44], Burkina [45], Cameroon [46], and Kenya [47]. However, the allelic frequency of the resistant allele recorded during this study was relatively low compared to those reported in Benin [48] and Cameroon [46] while being consistent with the previous record in Kenya [46]. In Benin and Cameroon, the high allele frequencies may be explained by the fixation of the mutation in the mosquito population, which is not the case in the population studied here. Furthermore, the presence of the *Gste2* mutation was correlated with the resistance to DDT in northern and southern Benin [44] in both *An. gambiae* and *Culex quinquefasciatus* populations from Benin [48]. Nevertheless, the resistant allele was likely selected during large-scale DDT use over the last decades [48]. Furthermore, the *Gste2* allele could have been selected by the increase of the selection pressure exerted by the recent extensive use of pyrethroids, which metabolized by *Gste2* enzyme. Pyrethroids are the unique chemical family recommended for use to impregnate LLINs, and their uncontrolled use in the agricultural sector was known in several areas [6,49].

The analysis of the sequencing data revealed high genetic and haplotypic diversities at *Ace1* and *Gste2* loci for the first time in Senegal using advanced sequencing approaches. The observed high genetic and haplotypic diversities at both loci suggest that they may also serve as a reservoir for new genetic variants at the early stage of the adaptive selection and evolution processes, as previously highlighted for *Ace-1* locus in *An. gambiae s.l.* [35] and the *Gste2* locus in the *Anopheles funestus* [40]. In contrast to this finding, low genetic diversity of the Gste2 gene was reported in *Anopheles funestus* in Benin [6]. However, additional genetic studies are needed to better understand the putative mechanisms underlying the *Ace1*- and *Gste2*-mediated resistances that could compromise the use of the carbamate and organophosphates as part of the insecticide management plan in vector control strategies in Senegal.

## 4. Material and Methods 

### 4.1. Study Area and Samples Collections

Historical samples of *An. gambiae s.l.* collected in 2013, 2017, and 2018 from ten entomological sentinel sites from Senegal (Figure 5) were used for this study. A random stratified sampling approach was used and drawn from an existing *An. gambiae s.l.* collection stored individually in silica gel at the Cheikh Anta Diop University to capture the spatial and temporal heterogeneities across the study area as much as possible. The selected health districts are located in the three main bioclimate zones accounting for almost 90% of the malaria transmission in Senegal [50].

### 4.2. Molecular Identification of An. gambiae s.l. Species and Genotyping of the Ace-1 and Gste2 Mutations

The genomic DNA was extracted from individuals using the DNeasy Blood & Tissue QIAcube kit following the manufacturer’s protocol. The *An. gambiae s.l.* sibling species were identified as described by Scott [51] and Fanelo [52]. The *Ace-1* and *Gste2* mutations were genotyped as previously described [6,53].

### 4.3. Partial Sequences of Ace-1 and Gste2 Genes 

An 817 bp fragment of the *Ace-1* gene was amplified using the Ex2Agdir1 (5′-AGGTCACGGTGAGTCCGTACGA-3′) and Ex4Agrev2 (5′-AGGGCGGACAGCAGATGCAGCGA-3′) pair of primers [39]. Moreover, a 748 bp fragment of *Gste2* was amplified using AgGste2F1 (5-GTACACCCTGCACCTTAGCC-3′) and AgGste2R1 (5-CCGTTCGCTTCCTCGTAGT-A-3′) pair of primers. The PCR conditions were set as described in Diallo et al. [50], with a slight modification. 

The PCR products were purified using Ampure xp beads (Beckman Coulter A63881) following the manufacturer’s guidelines. All the products were normalised to 20 ng, using 10 ng as a template (based on Thermofisher’s recommendations for fragment sizes between 500 bp–1 kb). Then, the forward and the reverse strands were amplified separately using the ready mix of the BigDye Terminator cycle sequencing kit V3.1. The sequencing reaction products were purified with Agencourt CleanSEQ (Ref A29151) before transferring to the ABI SeqStudio automatic sequencer (Applied Biosystems) for sequencing.

### 4.4. Data Analysis

The mutation frequencies were compared between sites and years using ANOVA at the significance level of 5%. The obtained sequences were confirmed using BioEdit v.7.2.1 and then aligned using the ClustalW [54]. Estimates of DNA polymorphism, including the number of segregating sites, number of haplotypes, haplotype diversity, nucleotide diversity, Tajima’s D, genetic differentiation, and gene flow between populations, were inferred using DnaSP v.5.10 [55]. The species relationship amongst haplotypes was estimated by constructing the haplotype network with Network v.10.2.0.0 software version [56], and the neighbour-joining phylogenetic tree was inferred using the MEGA v.7.0 program version [57] using the cleaned sequences aligned with reference sequences (accession numbers AGAP001356 for *Ace1* and AGAP009194 for *Gste2*) retrieved from VectorBase.

## 5. Conclusions

The present study is the first one highlighting the occurrence of the *Ace1* target-site resistance and *Gste2* metabolic mutation in the malaria vector population from Senegal. Although both mutations were found at low frequency, their spread is a serious concern for the future of the vector control strategies in Senegal. The evolution of insecticide resistance could negatively impact the effectiveness of all carbamate/organophosphates-based strategies, which are being used as a replacement for pyrethroid insecticides for managing the increasing resistances in the malaria vector across Senegal. Additional studies are required to better monitor the distribution and evolution of these two mutations and their association with the reported phenotypic resistance to preserve the limited insecticide-based vector control tools and ensure the sustainability of the gain achieved so far through the scale-up of proven-effective vector control strategies. However, the preliminary data generated from this study could be used by the NMPC to better guide and target vector control interventions across the country as part of their evidence-based vector control decision-making process.

## Figures and Tables

**Figure 1 pathogens-11-01021-f001:**
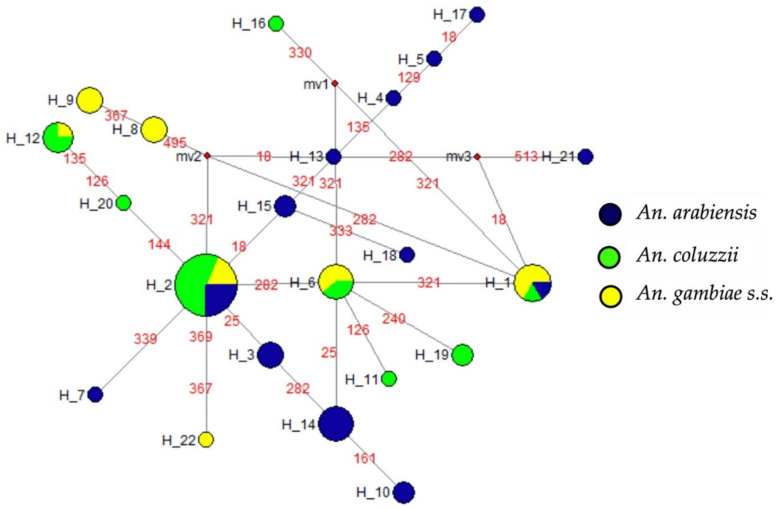
Haplotype diversity at *Ace-1* gene in *Anopheles gambiae s.l*. from Senegal. The circle size represents the frequency of the haplotype; the number in red corresponds to the position of the mutation; the small, red circles represent missing haplotypes that were not observed.

**Figure 2 pathogens-11-01021-f002:**
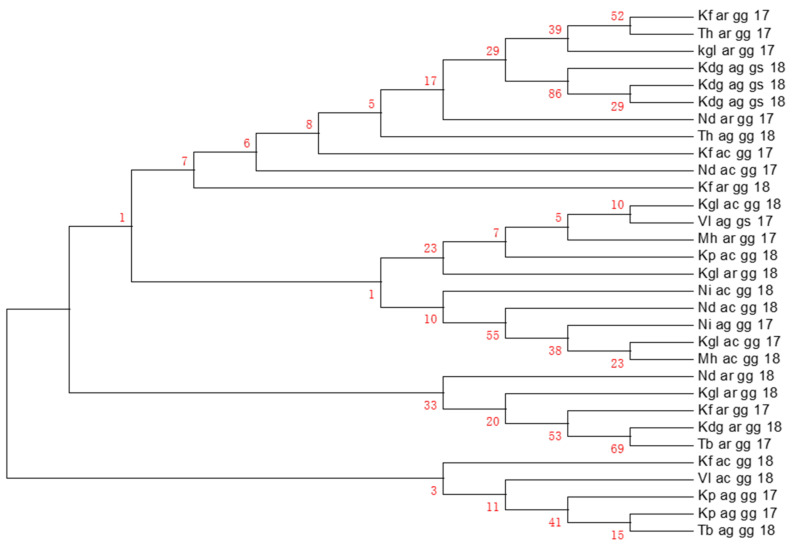
Evolutionary relationships of taxa using neighbour-joining method at *Ace-1 locus* between the *Anopheles gambiae s.l.* species from Senegal. The red numbers represent the confidence estimate value. **Study sites**: Kf, Kaffrine; TH, Thiés; Kgl, Koungheul; Kdg, Kédougou; Nd, Ndoffane; Vl, Vélingara; Mh, Malém Hoddar; Kp, Koumpentoum; NI, Nioro; Tb, Tambacounda. **Genotypes at *Ace-1 locus*:** gg, homozygote susceptible; gs, heterozygote resistance. **Species:** ar, ag, and ac correspond, respectively, to *An. arabiensis***,**
*An. gambiae s.s.*, and *An. coluzzii*. The numbers 13, 17, and 18 correspond to the study years (2013, 2017 and 2018 respectively).

**Figure 3 pathogens-11-01021-f003:**
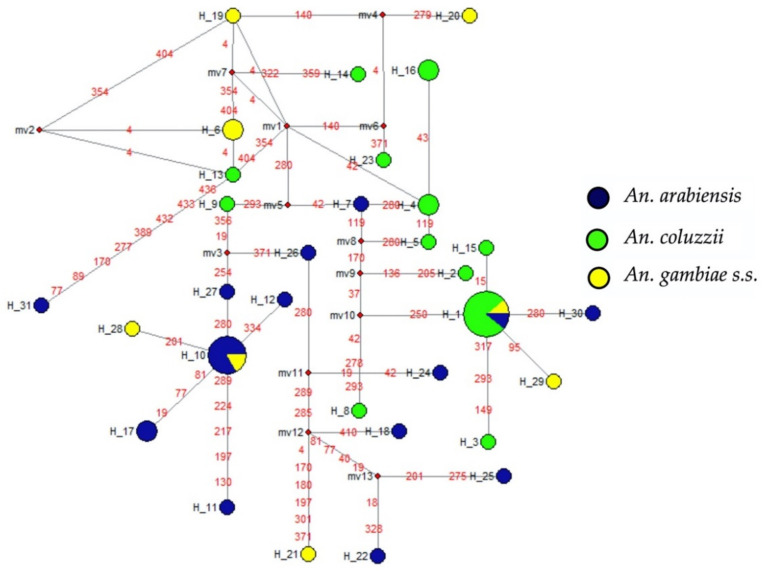
Haplotype diversity of Gste2 gene in *Anopheles gambiae s.l.* from Senegal. The circle size represents the haplotype frequency; the red number corresponds to the position of the mutations; the small, red circles represent missing haplotypes that were not observed.

**Figure 4 pathogens-11-01021-f004:**
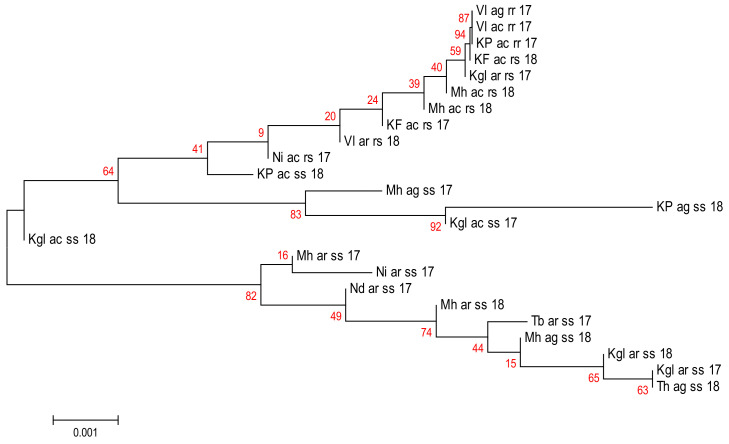
Evolutionary relationships of taxa using neighbour-joining method at *Gste2 locus* between the *Anopheles gambiae s.l.* species from Senegal. The red numbers represent the confidence estimate value. **Study sites**: Kf, Kaffrine; Th, Thiés; Kgl, Koungheul; Kdg, Kédougou; Nd, Ndoffane; Vl, Vélingara; Mh, Malém Hoddar; Kp, Koumpentoum; NI, Nioro; Tb, Tambacounda. **Genotypes at *Gste2 locus*:** gg, homozygote susceptible; gs, heterozygote resistance. **Species:** ar, ag, and ac correspond, respectively, to *An. arabiensis***,**
*An. gambiae s.s.*, and *An. coluzzii*. The numbers 13, 17, and 18 correspond to the study years (2013, 2017, and 2018, respectively).

**Figure 5 pathogens-11-01021-f005:**
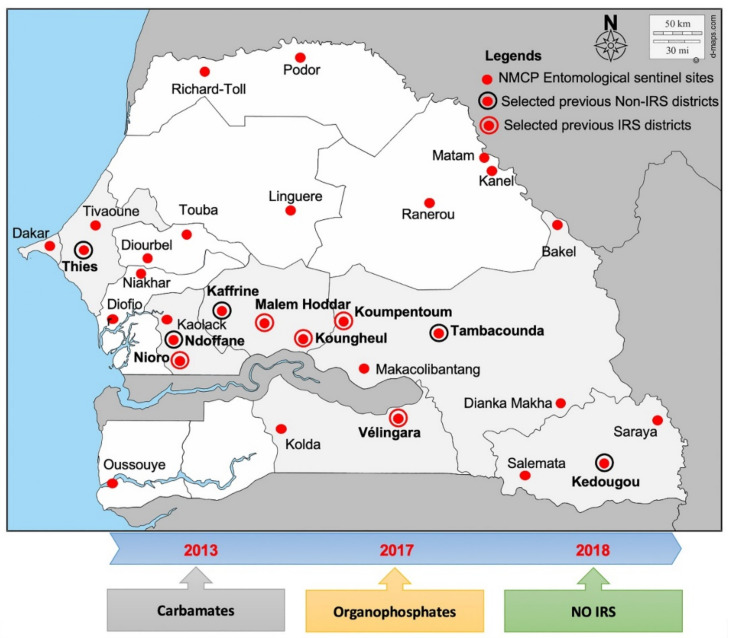
Geographical distribution of the study Health districts and insecticides used.

**Table 1 pathogens-11-01021-t001:** Genotypes and resistant allele frequencies of G119S-*Ace-1* mutation in natural populations of *Anopheles gambiae s.l.* from 10 Senegalese health districts.

			Genotypes				Frequency (S)	
Species	Locality	N	GG	GS	SS		G	S
**AR**	Kaffrine	87	87	0	0		1	0
	Kedougou	4	4	0	0		1	0
	Koumpentoum	89	89	0	0		1	0
	Koungheul	65	65	0	0		1	0
	Malem_Hodar	103	103	0	0		1	0
	Ndoffane	99	99	0	0		1	0
	Nioro	70	70	0	0		1	0
	Tambacounda	64	64	0	0		1	0
	Thies	113	113	0	0		1	0
	Velingara	54	54	0	0		1	0
	**Total**	**748**	**748**	**0**	**0**		**1**	**0**
**AC**	Kaffrine	16	16	0	0		1	0
	Kedougou	4	4	0	0		1	0
	Koumpentoum	14	14	0	0		1	0
	Koungheul	38	38	0	0		1	0
	Malem_Hodar	6	6	0	0		1	0
	Ndoffane	5	5	0	0		1	0
	Nioro	12	12	0	0		1	0
	Tambacounda	4	4	0	0		1	0
	Thies	3	3	0	0		1	0
	Velingara	11	11	0	0		1	0
	**Total**	**113**	**113**	**0**	**0**		**1**	**0**
**AG**	Kedougou	44	40	3	1		0.94	0.06
	Koumpentoum	6	6	0	0		1	0
	Malem_Hodar	1	1	0	0		1	0
	Nioro	19	19	0	0		1	0
	Tambacounda	35	33	2	0		0.97	0.03
	Thies	14	14	0	0		1	0
	Velingara	22	21	1	0		0.98	0.02
	**Total**	**141**	**134**	**6**	**1**		**0.97**	**0.03**
**AC-AG**	Nioro	2	2	0	0		1	0
	Tambacounda	5	4	1	0		0.9	0.1
	Thies	1	1	0	0		1	0
	Velingara	2	2	0	0		1	0
	**Total**	**10**	**9**	**1**	**0**		**0.95**	**0.05**

GG, glycine homozygous; SS, serine homozygous resistant; GS, glycine–serine heterozygous; AR, *An. arabiensis;* AG, *An. gambiae* s.s.; AC, *An. coluzzii*; AC-AG, *coluzzii-gambiae* hybrids; N, number of specimens tested per health district.

**Table 2 pathogens-11-01021-t002:** Genotypes and resistant alleles frequencies at the *Ace-1* gene in natural populations of *Anopheles gambiae s.l.* from three different sampling years.

			Genotypes				Frequency (S)	
Years	Species	N	GG	GS	SS		G	S
2013	**AR**	35	35	0	0		1	0
	**AC**	4	4	0	0		1	0
	**AG**	24	22	2	0		0.96	0.042
	**AC-AG**	4	3	1	0		0.88	0.125
	**Total**	**67**	**64**	**3**	**0**		**0.98**	**0.022**
2017	**AR**	348	348	0	0		1	0
	**AC**	43	43	0	0		1	0
	**AG**	37	36	1	0		0.99	0.01
	**AC-AG**	4	4	0	0		1	0
	**Total**	**432**	**431**	**1**	**0**		**0.999**	**0.001**
2018	**AR**	365	365	0	0		1	0
	**AC**	66	66	0	0		1	0
	**AG**	80	76	3	1		0.97	0.03
	**AC-AG**	2	2	0	0		1	0
	**Total**	**513**	**509**	**3**	**1**		**0.995**	**0.005**

**Table 3 pathogens-11-01021-t003:** Genotypes and resistant allele frequencies at *Gste2* in natural populations of *Anopheles gambiae s.l.* from 10 Senegalese health districts.

Species	Locality		Genotypes				Frequency (S)	
		N	RR	RS	SS		R	S
**AR**	Kaffrine	49	0	0	49		0	1
	Kedougou	3	0	0	3		0	1
	Koumpentoum	58	0	0	58		0	1
	Koungheul	40	0	0	40		0	1
	Malem_Hodar	58	0	0	58		0	1
	Ndoffane	37	1	0	36		0.03	0.97
	Nioro	32	0	0	32		0	1
	Tambacounda	40	1	0	39		0.03	0.97
	Thies	34	0	0	34		0	1
	Velingara	34	5	2	27		0.18	0.82
	**Total**	**385**	**7**	**2**	**376**		**0.02**	**0.98**
**AC**	Kaffrine	15	2	3	10		0.23	0.77
	Kedougou	4	0	2	2		0.25	0.75
	Koumpentoum	13	5	0	8		0.38	0.62
	Koungheul	36	5	0	31		0.14	0.86
	Malem_Hodar	7	1	1	5		0.21	0.79
	Ndoffane	2	0	0	2		0	1
	Nioro	11	3	0	8		0.27	0.73
	Tambacounda	3	0	0	3		0	1
	Thies	2	0	1	1		0.25	0.75
	Velingara	13	7	1	5		0.58	0.42
	**Total**	**106**	**23**	**8**	**75**		**0.25**	**0.75**
**AG**	Kedougou	53	7	3	43		0.16	0.84
	Koumpentoum	5	0	0	5		0	1
	Malem_Hodar	1	0	0	1		0	1
	Nioro	19	0	0	19		0	1
	Tambacounda	35	7	1	27		0.21	0.79
	Thies	7	0	0	7		0	1
	Velingara	14	1	1	12		0.11	0.89
	**Total**	**134**	**15**	**5**	**114**		**0.13**	**0.87**
**AC-AG**	Nioro	2	1	0	1		0.5	0.5
	Tambacounda	3	0	0	3		0	1
	Velingara	5	0	2	3		0.2	0.8
	**Total**	**10**	**1**	**2**	**7**		**0.2**	**0.8**

RR, homozygous resistant; SS, homozygous susceptible; RS, heterozygous; AR, *An. arabiensis;* AG, *An. gambiae* s.s.; AC, *An. coluzzii*; AC-AG, *coluzzii-gambiae* hybrids; N, number of specimens tested per health district.

**Table 4 pathogens-11-01021-t004:** Genotypes and resistant alleles frequencies at *Gste2* in natural populations of *Anopheles gambiae s.l.* from three different sampling years.

			Genotypes				Frequency (S)	
Years	Species	N	RR	RS	SS		R	S
2013	** *AR* **	17	0	0	17		0	1
	** *AC* **	4	0	1	3		0.13	0.88
	** *AG* **	23	5	1	17		0.24	0.76
	** *AC-AG* **	2	0	0	2		0	1
	**Total**	**46**	**5**	**2**	**39**		**0.13**	**0.87**
2017	** *AR* **	188	0	1	187		0.003	0.997
	** *AC* **	44	11	4	29		0.30	0.70
	** *AG* **	37	2	1	34		0.07	0.93
	** *AC-AG* **	7	0	2	5		0.14	0.86
	**Total**	**276**	**13**	**8**	**255**		**0.06**	**0.94**
2018	** *AR* **	180	7	1	172		0.04	0.96
	** *AC* **	58	12	3	43		0.23	0.77
	** *AG* **	74	8	3	63		0.13	0.87
	** *AC-AG* **	1	1		0		1	0
	**Total**	**313**	**28**	**7**	**278**		**0.10**	**0.90**

## Data Availability

Data supporting the conclusions of this article are included within the article. Raw data will be made available upon request to the corresponding author.

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
