# Peer review of "Evolution of the Ace-1 and Gste2 Mutations and Their Potential Impact on the Use of Carbamate and Organophosphates in IRS for Controlling Anopheles gambiae s.l., the Major Malaria Mosquito in Senegal"

_pathogens, 2022, doi:10.3390/pathogens11091021_

Round 1

Reviewer 1 Report

This manuscript describes screening of the Anopheles gambiae complex for two insecticide resistance mutations and describes the consequences for insecticide resistance management and vector control in Senegal. The results are important for this specific scenario.

I have detailed numerous editorial corrections in the attached file (please note that I converted the submitted pdf to a Word document so line numbers and other formatting features have become incorrect - please ignore this, but take notice of the editorial corrections made to the text).

Some specific areas to be addressed are as follows (please note that line numbers quoted refer to those in the pdf and not to those in the attached document):

Line 60 - Please write the name of the gene in full before using the abbreviation for the first time (for both Gste-2 and Ace-1)

Line 65 - Some explanation about the I114T mutation is required here to indicate that it enhances metabolic resistance rather than altering the target-site.

Line 72 - Use lower case for chemical groups

Line 74 - Please define ITNs before using the abbreviation. Elsewhere you refer to the nets as LLIN so perhaps you should just use this abbreviation here to be consistent (unless there is some other reason for using ITN.

Line 88 - Did any individual mosquito carry both Ace-1 and Gste2 mutations? This information should be mentioned and added to the Discussion.

Line 114 - It would be useful to see the presence and frequency of both mutations on a single map for each species so that spatial occurrence of the mutations could be visualised.

Line 266 - The possibility of gene duplication in Ace-1 should be described in the Introduction.

Line 267 - This sentence needs to be written more clearly so that we can understand what 'no known mix nucleotide position' means and what you would have expected to see if there were gene duplication at this locus.

Line 317 - Which Qiagen kit was used?

Line 327 - What was the modification and why was it necessary? Others may like to use your protocol, so they must be told what it was.

In the References, please check that genus and species are italicized. Remove capital letter from specific names.

Author Response

Reviewer 1 comment 1: Line 60 - Please write the name of the gene in full before using the abbreviation for the first time (for both Gste-2 and Ace-1)

Reviewer 1 Answer 1: Corrected.

Reviewer 1 comment 2: Line 65 - Some explanation about the I114T mutation is required here to indicate that it enhances metabolic resistance rather than altering the target-site.

Reviewer 1 Answer 2: Corrected as suggested (see Line 66).

Reviewer 1 comment 3: Line 72 - Use lower case for chemical groups

Reviewer 1 Answer 3: Corrected.

Reviewer 1 comment 4: Line 74 - Please define ITNs before using the abbreviation. Elsewhere you refer to the nets as LLIN so perhaps you should just use this abbreviation here to be consistent (unless there is some other reason for using ITN.

Reviewer 1 Answer 4: Corrected as suggested.

Reviewer 1 comment 5: Line 88 - Did any individual mosquito carry both Ace-1 and Gste2 mutations? This information should be mentioned and added to the Discussion.

Reviewer 1 Answer 5: There was only one sample carrying the both mutations at heterozygote stage. We judged unnecessary to speculate on this because it frequency was really low.

Reviewer 1 comment 6: Line 114 - It would be useful to see the presence and frequency of both mutations on a single map for each species so that spatial occurrence of the mutations could be visualised.

Reviewer 1 Answer 6: we thank the reviewer for this suggestion. The frequency of both mutations was already presented in tables 1 and 3 and it may be redundant to include it on a new map. However, we can consider this visual aspect in our future data presentation.

Reviewer 1 comment 7: Line 266 - The possibility of gene duplication in Ace-1 should be described in the Introduction.

Reviewer 1 Answer 7: Included as suggested (see line 72-47)

Reviewer 1 comment 8: Line 267 - This sentence needs to be written more clearly so that we can understand what 'no known mix nucleotide position' means and what you would have expected to see if there were gene duplication at this locus.

Reviewer 1 Answer 8: Corrected accordingly. We hope that it is better now.

Reviewer 1 comment 9: Line 317 - Which Qiagen kit was used?

Reviewer 1 Answer 9: We used DNeasy Blood & Tissue QIAcube kit. We included this in the new version of the manuscript (see Line 321).

Reviewer 1 comment 10: Line 327 - What was the modification and why was it necessary? Others may like to use your protocol, so they must be told what it was.

Reviewer 1 Answer 10: The slight modification was the hybridization temperature of the specific primers targeting Ace-1 and Gste2 genes.

Reviewer 1 comment 11: In the References, please check that genus and species are italicized. Remove capital letter from specific names.

Reviewer 1 Answer 11: Corrected.

Reviewer 2 Report

The manuscript entitled “ Evolution of the Ace-1 and Gste2 mutations and their potential 2 impact on the use of carbamate and organophosphates in IRS 3 for controlling Anopheles gambiae s.l., the major malaria mos-4 quito in Senegal ” by Diallo et al. reported the presence and evolution of the Ace-1 and Gste2 resistance genes in natural populations of Anopheles gambiae The reported findings are useful for supporting the NMCP for evidence based vector control interventions selection and targeting.

The manuscript was well prepared. No major faults were found except that some minor errors in format and spellings.

Minors

Lines 62-65: “Moreover, previous studies have reported 5–8 fold Gste2 over-expression in An. gambiae display ing phenotypic DDT resistance while the voltage-gated sodium channel mutation was not detected [7][8]. “ Here, the font size of Gste2 and An. gambiae loos bigger than others. Also check others through out the text.

Figure 4: In the legend, the red numbers and scale in the figure should be described. Similar corrections should also made in other figures in order to be understood easier.

Ace-1 and Gste2: I did not find the complete names for both abbreviations. Authors may think to provide full names for both genes for most audience to follow when they first appear in the paper.

Author Response

Reviewer 2 comment 1: Lines 62-65: “Moreover, previous studies have reported 5–8 fold Gste2 over-expression in An. gambiae displaying phenotypic DDT resistance while the voltage-gated sodium channel mutation was not detected [7][8]. “ Here, the font size of Gste2 and An. gambiae loos bigger than others. Also check others throughout the text.

Reviewer 2 Answer 1: Corrected as suggested.

Reviewer 2 comment 2: Figure 4: In the legend, the red numbers and scale in the figure should be described. Similar corrections should also made in other figures in order to be understood easier.

 Reviewer 2 Answer 2: Corrected as suggested.

Reviewer 2 comment 3: Ace-1 and Gste2: I did not find the complete names for both abbreviations. Authors may think to provide full names for both genes for the most audience to follow when they first appear in the paper.

Reviewer 2 Answer 3: Corrected as suggested.
